# Effectiveness of routine tuberculosis education in a high-burden setting: A prospective observational cohort study

Tyler Scott Johnson[1,2], Leah Nanziri[3], Amanda J. Gupta[1,2,3], Joseph M. Ggita[3], Mari Armstrong-Hough[3,4], Irene Ayakaka[3], Sheela V. Shenoi[5], Achilles Katamba[3,6], J. Lucian Davis[1,3,7,8]*

1 Department of Epidemiology of Microbial Diseases, Yale School of Public Health, New Haven, Connecticut, United States of America, 2 Johns Hopkins Bloomberg School of Public Health, Baltimore, Maryland, United States of America, 3 World Alliance for Lung and Intensive Care Medicine in Uganda (WALIMU), Kampala, Uganda, 4 Department of Social and Behavioral Sciences, School of Global Public Health, New York University, New York, New York, United States of America, 5 AIDS Program, Section of Infectious Diseases, Yale School of Medicine, New Haven, Connecticut, United States of America, 6 Clinical Epidemiology Unit, Makerere University, Kampala, Uganda, 7 Center for Methods in Implementation and Prevention Science, Yale School of Public Health, New Haven, Connecticut, United States of America, 8 Section of Pulmonary, Critical Care, and Sleep Medicine, Yale School of Medicine, New Haven, Connecticut, United States of America

* lucian.davis@yale.edu

## Abstract

### Background

Low adherence to tuberculosis (TB) treatment remains a major driver of adverse health outcomes in high-burden countries. While guidelines recommend routine client education, its effectiveness and the knowledge constructs that influence adherence remain poorly defined.

### Methods

We conducted a prospective cohort study of adults (≥18 years) initiating treatment for drug-susceptible TB in Kampala, Uganda. We assessed clients' TB knowledge before and after routine education and at the two-week, two-month, and five-month follow-up visits. We recorded self-reported seven-day medication adherence at each visit and final treatment outcomes. We used paired t-tests to compare knowledge scores (0–100) before and after education. We constructed multivariable Poisson and logistic regression models to examine the association of knowledge with the outcomes of nonadherence and WHO-defined treatment success.

### Results

We enrolled 80 participants (28% female; 36% living with HIV). Overall TB knowledge scores increased by 25 points (95% CI 22–28), from 53 to 78, yet only 9% achieved TB literacy (an overall TB knowledge score ≥90) after TB education, peaking at 18%

**Data availability statement:** All relevant data, including the de-identified dataset, README file, variable labels, and R analysis code, are available from the Yale Dataverse repository at the following URL: https://doi.org/10.60600/YU/HKO6AJ. The citation is as follows: Johnson, T. S., Nanziri, L., Gupta, A. J., Armstrong-Hough, M., Ayakaka, I., Shenoi, S. V., Katamba, A., & Davis, J. L. (2026). Replication Data for: Effectiveness of routine tuberculosis education in a high-burden setting: A prospective observational cohort study. Yale Dataverse, V1. doi.org.

**Funding:** This research was supported by the National Institute of Mental Health of the National Institutes of Health through a pilot award to JLD from the Yale Center for Interdisciplinary Research on AIDS under award number P30MH062294 and by the National Heart Lung and Blood Institute under award number R01HL176337 (JLD). In addition, Yale University's Wilbur G. Downs Fellowship (TJ) and Lindsay Fellowship for Research in Africa (TJ) also supported this work. The content is solely the responsibility of the authors and does not necessarily represent the official views of the National Institutes of Health or other research funders. The funders had no role in study design, data collection and analysis, decision to publish, or preparation of the manuscript.

**Competing interests:** The authors have declared that no competing interests exist.

at two weeks, before declining to just 3.6% at five months. Each 10-point increase in post-education knowledge was associated with a reduction in nonadherence of 15% ($IRR^{0.1}$ 0.85; 95% CI 0.72–1.00, p = 0.048) at the two-week visit and 32% ($IRR^{0.1}$ 0.68; 95% CI 0.50–0.89, p = 0.005) at the two-month visit. Ultimately, 71% completed treatment, but knowledge was not a significant predictor of treatment success (OR = 1.77, 95% CI 0.01–314.6, p = 0.83).

## Conclusions

Routine client education significantly increased TB knowledge, yet less than 10% achieved TB literacy. Post-education knowledge was independently associated with better early adherence, highlighting the need for more robust education interventions to optimize TB treatment outcomes.

---

## Introduction

A striking paradox of the global tuberculosis (TB) epidemic is that 7–29% of adults worldwide still experience adverse outcomes, including death, failure, loss-to-follow-up, and relapse [1–4], despite the widespread availability of safe and effective medications for drug-susceptible TB disease. Low adherence to TB treatment remains a major driver of these poor outcomes and contributes to antimicrobial resistance [5,6].

To improve TB medication adherence, WHO guidelines recommend providing education to all individuals starting TB treatment as a core component of person-centered TB management [4,7]. Furthermore, the Patient's Charter for TB Care states that attaining a high level of treatment literacy is a fundamental right of all persons with TB [8]. Client education aims to increase knowledge about TB disease and treatment [9]; however, current evidence indicates that it improves TB medication adherence and treatment outcomes primarily when combined with additional targeted interventions, such as reminders and incentives [9–13]. While previous studies have shown that TB knowledge influences treatment outcomes, the specific constructs of TB knowledge driving this association remain unclear [14–16]. Moreover, very few studies have assessed the independent effect of client education on medication adherence, leaving key pathways between education and treatment outcomes unexplored [17]. Finally, little is known about the quality and content of TB education as delivered in routine practice in resource-constrained settings. Therefore, we sought to evaluate the effectiveness of routine client education in improving knowledge about TB disease and treatment and to identify which aspects of knowledge are associated with medication adherence among persons with TB in a resource-constrained setting in Kampala, Uganda.

## Materials and methods

### Study objectives

Because terminology lacks specificity across guidelines and the published literature, we operationally defined TB education as structured interventions designed to shape clients' knowledge, attitudes, and practices by explaining the disease and its

treatment. In contrast, we defined counseling as individualized support focused on managing TB treatment and resolving challenges to adherence and well-being such as stigma, concerns about side effects, and low self-efficacy [17,18]. Because there is limited published guidance on TB counseling in Uganda, this study focuses on the routine TB education provided to clients at TB treatment initiation. We had four objectives for our analysis: [1] to determine the effectiveness of client education in increasing TB knowledge among individuals newly diagnosed with TB disease; [2] to evaluate the retention of TB knowledge throughout treatment; [3] to examine the association between post-education knowledge and treatment nonadherence; and [4] to assess whether TB knowledge is associated with treatment outcomes.

## Study design and setting

We conducted a prospective cohort study to evaluate the effectiveness of routine client education on TB knowledge and its impact on treatment adherence and outcomes among adults starting TB treatment at Kisenyi Health Centre in Kampala, Uganda. This large, urban, public primary health clinic offers free TB diagnostic and treatment services through the Uganda National TB and Leprosy Programme (NTLP).

In Uganda, TB clients are typically scheduled to return to the clinic every two weeks for medication refills during the first two months of treatment, and monthly for the remaining four months. National TB guidelines recommend that healthcare workers provide TB education to all individuals diagnosed with TB at the time of diagnosis and throughout the six-month treatment period. However, in practice, education is usually provided only at the initial visit. The Uganda Ministry of Health offers a TB and HIV Health Education Flipchart that covers key biomedical concepts using images and descriptive text in English and Luganda, the predominant languages spoken in Kampala (S1 File) [19]. The content and delivery of TB education lack standardization and are primarily left to individual staff. At Kisenyi, TB unit nurses and assistant nurses provide client education individually or in groups, sometimes using the flipchart to illustrate key concepts.

## Development of an instrument for assessing TB knowledge

We developed a new survey instrument to measure TB knowledge (S1 Appendix), encompassing eight constructs related to TB disease and treatment: TB basics, TB symptoms, TB transmission, HIV-TB interactions, TB prevention, TB treatment principles, TB treatment regimen, and TB treatment follow-up. We defined these constructs based on topics included in the Uganda TB education flipchart and in counseling guidelines published by Médecins Sans Frontières [19,20]. We further drew on established education and counseling content and principles used to support HIV medication adherence and aligned the eight constructs with guidance in the WHO operational handbook on TB [21,22]. To establish face validity, members of the Ugandan and U.S. research teams with expertise in TB care and research reviewed the survey and provided feedback. We then piloted the questionnaire with 12 individuals undergoing TB treatment to assess its feasibility and acceptability before starting the study.

## Participants

We enrolled consecutive adults (aged ≥18 years) initiating TB treatment as documented in the Uganda National TB and Leprosy Programme Treatment Register. We excluded individuals undergoing retreatment, those with drug-resistant TB, clients referred after receiving TB education, individuals transferring in or planning to transfer out to another facility, and those who declined or lacked the capacity to provide informed consent.

## Procedures

A research officer (L.N.) collected clinical and demographic data and verbally administered a baseline TB knowledge assessment to study participants before they received any education (S1 Appendix). We recorded whether education was delivered individually or in a group and whether the NTLP-endorsed flipchart was used. The research officer repeated the TB knowledge assessment after the education session and again at the scheduled two-week, two-month, and five-month

treatment monitoring and medication refill visits (**Fig 1**). At each follow-up visit, we recorded self-reported adherence to TB medications using a validated seven-day recall measure. This method has been shown to correlate well with directly observed adherence measures and to predict clinical outcomes of antiretroviral therapy for HIV [23]. The research officer recorded final treatment outcomes at eight months by auditing the TB treatment register and applying standardized WHO and NTLP definitions [4,24]. We collected all survey data electronically using a customized Qualtrics (Provo, Utah, USA) survey application.

## Outcome measures

To score the knowledge instrument, we calculated the proportion of correct responses to questions related to each construct for each participant at each time point to create eight construct-specific scores (individual question scores included in S2 Appendix). We then averaged these construct scores to create an overall TB knowledge score and standardized the resulting value on a 0–100 scale. To enhance interpretability, we defined a post-hoc secondary outcome, TB literacy, operationalized as an overall TB knowledge score ≥90 points. We selected this high threshold for TB literacy based on the foundational knowledge assessed by the questions and the strong emphasis on receiving and sharing information about TB in the Patient's Charter for TB Care. We defined treatment nonadherence as the reported proportion of doses missed or taken late, calculated from responses to the seven-day recall measure.

## Statistical analysis

We summarized clinical and demographic characteristics, educational exposures, and TB literacy using simple proportions and TB knowledge scores using means and standard deviations. We assessed changes in TB knowledge after baseline client education and evaluated knowledge retention at each follow-up visit using two-sided paired t-tests overall and at the construct level. We fit multivariate Poisson regression models to examine the association between post-education knowledge scores and TB treatment nonadherence, expressed as incidence rate ratios (IRRs). Analyses were restricted to participants who attended follow-up visits where self-reported nonadherence data were collected. We adjusted all models

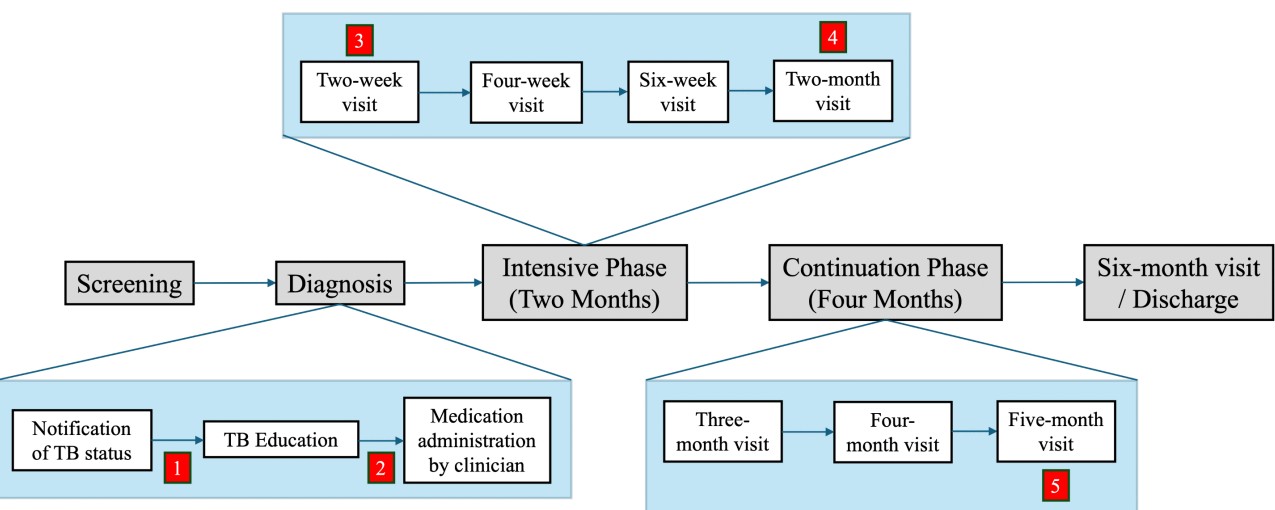

**Fig 1. Timeline of the TB treatment pathway showing predefined study measurement time points.** The figure illustrates the clinical pathway from screening and diagnosis through the intensive phase (first two months) and continuation phase (subsequent four months) of TB treatment, ending at the six-month visit and discharge. Boxes depict routine clinical steps and scheduled follow-up visits during treatment. Red numbered markers indicate the predefined study measurement time points. Abbreviations: tuberculosis (TB).

for factors previously associated with treatment outcomes, including sex, HIV status, education level, prior TB history, and type of education received (i.e., individual vs. group) [5]. To ease interpretation, we expressed effects estimates per 10-unit increase in knowledge score by transforming the model IRR to the power of 0.10. We calculated dispersion ratios for all Poisson models. To account for mild overdispersion, we fit all Poisson regression models using robust (sandwich) standard errors.

We used bivariate logistic regression models to evaluate the association between post-education knowledge scores and WHO-defined TB treatment outcomes. [25] We constructed a multivariate model using backward stepwise selection, removing variables with p-values ≥0.2. We included age, sex, HIV status, and education level in the model for face validity. We defined statistical tests as significant at the $p \le 0.05$ level. To assess for potential attrition bias, we compared participants lost to follow-up with those with a recorded final treatment outcome using Fisher's exact or the Wilcoxon rank-sum test, as appropriate. All statistical analyses were conducted using R version 4.3.2 (R Core Team, 2023).

### Sample size estimates

We planned to enroll 80 adults with pulmonary TB. We conducted power calculations for each objective using the total TB knowledge score (0–100). For Objectives 1 and 2, we assumed a baseline mean score of 50 (standard deviation 20) based on prior studies [26–29]. With an expected 30-point increase and a within-person correlation of 0.75, we estimated that a sample of 70 participants would provide 99% power at α = 0.05. For Objective 3, assuming a baseline of three missed or late doses per week, we estimated that 75 participants would provide 88% power to detect a 10% reduction in nonadherence per ten-point increase in overall knowledge score using a Poisson regression model. For Objective 4, we estimated that detecting a 15% increase in treatment success to 85% from a baseline of 70% would require 70 participants to achieve 93% power. [30]. Because the adherence objective required the largest sample size, we set the final target sample size as 80 participants, to allow for a modest loss of participants to follow-up.

### Human subjects' protection

Data collection began on June 5, 2018, and ended on January 16, 2019. All participants provided written informed consent. The Makerere College of Health Sciences Institutional Review Board, the Uganda National Council for Science and Technology, and the Yale University Human Investigation Committee approved the study protocol.

## Results

We enrolled 80 persons with newly diagnosed TB disease from June to August 2018 (Fig 2). Twenty-two (28%) were female, 29 (36%) were persons living with HIV (PLWH), and 24 (30%) had one or more prior episodes of TB disease (Table 1). Group education was provided to 43 clients (54%), while the remaining 37 (46%) clients received individual education. The NTLP-endorsed flipchart was used to educate 45 clients (56%). We excluded two individuals from follow-up analyses because they transferred their care to other treatment centers after the initial visit.

### Baseline TB knowledge and effect of client education on TB knowledge

The mean overall TB Knowledge score at baseline was 53 points (standard deviation, SD 13), which increased to 78 points (SD 11) after education (+25 points, 95% CI 22–28, Table 2). Baseline scores ranged from 20 points (SD 29) for the TB Treatment Follow-up construct to 77 points (SD 24) for the HIV-TB Relationship construct. After receiving TB education, construct scores ranged from 65 points (SD 24) for TB Basics to 96 points (SD 8.1) for TB Treatment Principles. There were statistically significant within-subject increases in knowledge scores across all eight constructs. The largest improvement was in the TB Treatment Follow-up construct (+54 points, 95% CI 47–62), while the smallest was in the TB Symptoms construct (+9 points, 95% CI 3–14).

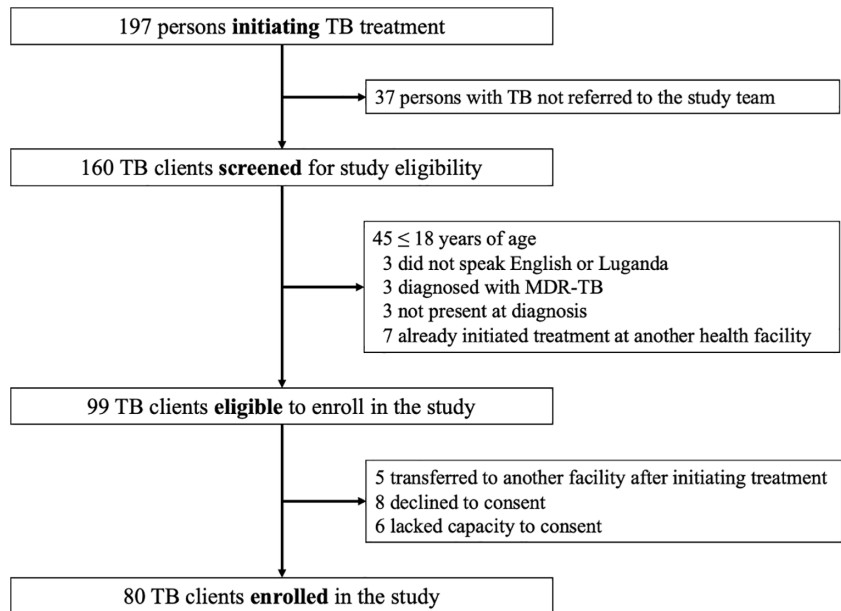

**Fig 2. Flow diagram showing participant screening and enrollment.** Abbreviations: tuberculosis (TB); multi-drug resistant TB (MDR-TB).

Overall, only 9% of participants achieved post-education scores ≥90 for their overall TB knowledge score, our benchmark for TB literacy and high-quality client education (**Fig 3**). At the construct level, the proportions of clients achieving TB literacy after baseline education were 23% for TB Basics, 23% for TB Treatment Regimen, 33% for TB symptoms, 25% for TB Transmission, 51% for TB Treatment Follow-up, 56% for HIV-TB Relationship, 80% for TB Treatment Principles, and 83% for TB Prevention.

### TB knowledge retention after diagnosis

This proportion achieving TB literacy increased to 18% at two weeks, before declining to just 3.6% at 5 months (**Fig 3**). Patterns of literacy retention across individual constructs were more variable. For some constructs, the proportion of participants with TB literacy increased over time (TB Treatment Principles, HIV-TB Relationship), whereas for others (TB Basics, TB Transmission, TB Treatment Follow-up, TB Treatment Regimen), this proportion decreased over time. Literacy for the remaining constructs (TB Symptoms, TB Prevention) remained relatively stable over time.

### TB knowledge and treatment outcomes

The mean number of days with missed or late doses in the preceding seven days was 2.1 at two weeks (SD 1.9), 1.2 at two months (SD 1.6), and 0.78 at five months (SD 1.5). In multivariable Poisson models adjusted for sex, HIV status, education level, prior TB history, and modality of education, higher post-education overall TB knowledge was associated with lower nonadherence early in treatment. At the two-week visit, each 10-point increase in the overall knowledge score was associated with an estimated 15% reduction in missed or late doses ($IRR^{0.1} = 0.85$, 95% CI 0.72–1.00, $p = 0.048$), and, at the two-month visit, with a 32% reduction ($IRR^{0.1} = 0.68$, 95% CI 0.50–0.89, $p = 0.005$) (**Table 3**). By the five-month visit, the association between post-education knowledge and nonadherence was no longer statistically significant ($IRR^{0.1} = 0.89$, 95% CI 0.62–1.28, $p = 0.52$).

**Table 1. Baseline characteristics of study participants.**

| Characteristic (n = 80) | Frequency | Percentage* |
|---|---|---|
| Sex | | |
| Male | 58 | 73% |
| Female | 22 | 28% |
| Age (years) | | |
| 18-29 | 38 | 48% |
| 30-39 | 23 | 29% |
| 40-49 | 16 | 20% |
| ≥50 | 3 | 4% |
| Education | | |
| Did not complete primary school | 30 | 37% |
| Completed primary school | 50 | 63% |
| HIV Status | | |
| Living with HIV | 29 | 36% |
| Not living with HIV | 51 | 64% |
| Prior history of TB | | |
| Yes | 24 | 30% |
| No | 56 | 70% |
| Type of client education | | |
| Group | 43 | 54% |
| Individual | 37 | 46% |
| Flipchart used for client education | | |
| Yes | 45 | 56% |
| No | 35 | 44% |
| TB Diagnosis | | |
| Pulmonary, bacteriologically confirmed | 58 | 73% |
| Pulmonary, clinically diagnosed | 17 | 21% |
| Extrapulmonary | 4 | 5% |
| Not recorded | 1 | 1% |

**Abbreviations:** CI, Confidence Interval; TB, tuberculosis

**Legend:** *Percentages may exceed 100% due to rounding

By the end of the eight-month follow-up period, 55 of 78 eligible individuals (71%) had completed TB treatment, while 23 (29%) had not, including two (2.6%) who had died (both PLWH), four (5.1%) who had experienced treatment failure, and 17 (22%) who were lost to follow-up. Compared with participants who remained in care, those lost to follow-up did not differ significantly by sex, HIV status, education level, prior TB history, or education modality (results not shown). In multivariable logistic models adjusting for demographic and clinical covariates, the overall post-education knowledge score was not significantly associated with treatment success (OR = 1.77, 95% CI 0.01–314.6, p = 0.83) (**Table 3**). The wide confidence interval reflects the modest sample size and small number of unfavorable outcomes.

## Discussion

Education is universally recommended by both the WHO and the Uganda NTLP for individuals initiating TB treatment, based on stakeholder values and preferences and supported by low-to-moderate-quality evidence supporting its effectiveness in improving treatment outcomes [4,8,9,24]. Routine client education was associated with significant increases

**Table 2. TB knowledge scores before and immediately after routine client education (n = 80), with constructs ordered by the magnitude of change in scores.**

| Content | Pre-education | Post-education | Change[1] | 95% CI | p-value[2] |
|---|---|---|---|---|---|
| | **Score** (SD) | **Score** (SD) | % | | |
| *Overall TB Knowledge* | 53 (13) | 78 (11) | +25 | 22-28 | <0.0001 |
| **Constructs[1]** | | | | | |
| *TB Treatment Follow-up* | 20 (29) | 74 (28) | +54 | 47-62 | <0.0001 |
| *TB Treatment Regimen* | 32 (22) | 74 (22) | +42 | 37-47 | <0.0001 |
| *TB Prevention* | 47 (50) | 83 (38) | +35 | 22-48 | <0.0001 |
| *TB Treatment Principles* | 71 (19) | 96 (8.1) | +25 | 20-29 | <0.0001 |
| *TB Basics* | 49 (23) | 65 (24) | +16 | 8-23 | 0.0001 |
| *TB Transmission* | 54 (27) | 69 (23) | +14 | 8-21 | <0.0001 |
| *TB Symptoms* | 71 (23) | 80 (18) | +9 | 3-14 | 0.002 |
| *HIV-TB Relationship* | 77 (24) | 85 (18) | +7 | 2-13 | 0.009 |

**Abbreviations:** CI, Confidence Interval; SD, standard deviation; TB, tuberculosis.

**Legend:** [1]Change in Score" values for each construct represent the average within-participant differences in the sample (i.e., each individual's post-education score minus their pre-education score, averaged across all participants).

3 p-value tests the null hypothesis that the true population mean difference is zero.

in TB knowledge, which further increased during the early intensive phase of TB treatment. However, standard TB education appeared to be insufficient, with the proportion achieving high levels of knowledge associated with overall TB literacy never exceeding 20% at any time point and falling below 5% by the end of treatment. Higher knowledge levels were associated with improved adherence early in treatment, but this association diminished over time after the initial education session.

Previous studies have also documented low levels of TB literacy among clients and community members. A cross-sectional survey among residents of three urban slums in Kampala revealed low baseline TB knowledge in a similar study population, underscoring the need for robust client education at the time of clinic presentation and evaluation for TB disease [31]. Similar findings have been reported elsewhere. In South Africa, one study found that although general awareness of TB was high, significant misconceptions about transmission and treatment duration persisted, contributing to delays in care-seeking [32]. A broader systematic review similarly reported wide variation in TB knowledge across settings, with only 0–63% of individuals demonstrating an accurate understanding of the causes of TB. The review also noted frequent confusion among clients between TB and HIV, which fueled stigma and impeded engagement with care [33]. Our analysis builds on these findings by showing that client education can effectively address baseline knowledge gaps while also helping identify specific areas that remain inadequately addressed. For example, our cohort showed a stronger command of constructs related to treatment than those related to the disease itself, suggesting that routine education emphasizes the biomedical aspects of treatment more than explanations of the disease. Future client education interventions should be longitudinal and guided by assessments of both baseline and post-education TB knowledge to better identify persistent literacy gaps and better determine where routine education is falling short.

A major barrier to advancing research on TB literacy is the lack of standardized, validated measurement instruments. Many studies rely on custom-designed tools without rigorous validation, limiting the comparability and reproducibility of findings across settings [34]. For this study, we developed a novel, content-valid instrument covering eight constructs universally recommended for TB treatment education and adapted to an urban Ugandan context. This tool was reviewed by TB experts in Uganda and the U.S., piloted for feasibility, and applied longitudinally to capture within-subject changes, thereby enhancing the rigor, reproducibility, and contextual relevance of our findings.

## Overall TB Knowledge

## Knowledge Constructs

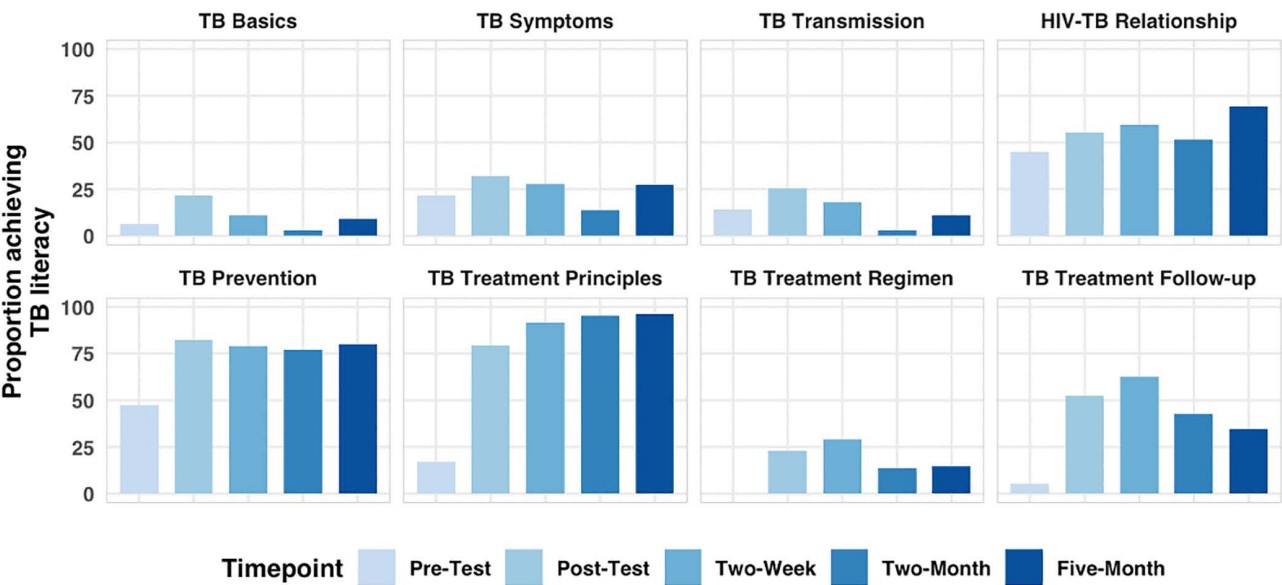

**Fig 3. Proportion of participants achieving TB literacy overall and by construct across all time points.** A 30-item questionnaire was administered before and after routine education and at two-week, two-month, and five-month follow-up visits. Items were grouped into eight constructs and combined into an overall knowledge score. The figure displays the proportion of participants who achieved TB literacy (defined as a knowledge score ≥90) for the overall TB knowledge (top panels) and for each of the knowledge constructs (bottom panels) at every time point. Abbreviations: tuberculosis (TB).

Nonadherence is a well-established predictor of poor treatment outcomes. A pooled analysis of the control arms of three clinical trials found that missing just 10% of doses was the strongest predictor of unfavorable TB outcomes [35]. Our finding that knowledge was significantly associated with lower nonadherence early in treatment mirrors similar associations observed in prior facility-based studies, which found that inadequate TB knowledge contributes to delays in treatment initiation and increased risk of loss to follow-up. For example, a cross-sectional study in Nigeria showed that low TB symptom knowledge was associated with delayed treatment-seeking behavior [36]. In addition, a case-control study in Morocco found that greater TB knowledge was associated with better medication adherence [16]. These findings suggest that even modest gains in TB knowledge could lead to meaningful improvements in adherence and retention, making enhanced client education a promising intervention for improving treatment outcomes.

Recommendations for providing TB education throughout treatment are already incorporated into global and national guidelines, lending immediate programmatic relevance to our findings, but more work is needed on strategies for

**Table 3. Adjusted associations between post-education knowledge scores and self-reported nonadherence and final treatment outcome.**

| Predictor | Week 2 Nonadherence (n=73*) | | | Month 2 Nonadherence (n=67*) | | | Month 5 Nonadherence (n=57*) | | | Treatment Success (n=78*) | | |
|---|---|---|---|---|---|---|---|---|---|---|---|---|
| | $IRR^{0.1}$ | 95% CI | p-value | $IRR^{0.1}$ | 95% CI | p-value | $IRR^{0.1}$ | 95% CI | p-value | OR | 95% CI | p-value |
| Post-Education TB Knowledge | 0.85 | 0.72–1.00 | 0.048 | 0.68 | 0.50–0.89 | 0.005 | 0.89 | 0.62–1.28 | 0.52 | 1.77 | 0.01–314.6 | 0.83 |
| Sex | 1.00 | 0.96–1.04 | 0.83 | 0.99 | 0.92–1.06 | 0.78 | 1.06 | 0.92–1.21 | 0.42 | 2.09 | 0.63–8.36 | 0.25 |
| HIV Status | 1.03 | 0.99–1.07 | 0.19 | 1.00 | 0.94–1.06 | 0.96 | 1.02 | 0.91–1.13 | 0.78 | 0.65 | 0.21–1.95 | 0.43 |
| Completed Primary School | 1.00 | 0.96–1.05 | 0.87 | 1.07 | 1.00–1.15 | 0.05 | 1.03 | 0.90–1.19 | 0.64 | 0.83 | 0.25–2.60 | 0.75 |
| Prior TB History | 0.97 | 0.93–1.02 | 0.31 | 0.93 | 0.85–1.01 | 0.06 | 0.99 | 0.89–1.10 | 0.82 | 2.58 | 0.76–10.65 | 0.15 |
| Individual Education | 0.98 | 0.94–1.02 | 0.38 | 1.04 | 0.98–1.10 | 0.20 | 0.98 | 0.87–1.11 | 0.75 | 2.80 | 0.97–8.87 | 0.07 |

**Abbreviations:** CI, confidence interval; IRR, incidence rate ratio; OR, odds ratio; TB, tuberculosis.

**Legend:** *We excluded from nonadherence analyses participants who were lost to follow-up before their corresponding medication refill appointments at two weeks, two months, and five months. Of 80 subjects initially enrolled, we excluded two individuals who transferred to other treatment centers from all analyses.

real-world implementation. In a previous qualitative study that we conducted in the same clinic, people with TB and healthcare workers both recommended a more structured approach to TB education, including task-sharing of education with trained lay providers or CHWs [18]. In many settings, this education could be delivered during existing waiting periods before clinical encounters and supported by peer-navigation models that have demonstrated effectiveness in similar contexts [37,38].

Our study had several notable strengths. The paired, pre-post survey design, administered at diagnosis and throughout treatment, enabled within-individual analysis of knowledge gains using paired t-tests, increased statistical power, and allowed for a granular assessment of knowledge retention throughout the intensive and continuation phases of treatment. Furthermore, introducing a standardized assessment of TB disease and treatment knowledge for individuals initiating TB treatment adds a valuable tool to the literature. This knowledge assessment tool may facilitate future studies and support cross-study comparisons.

Our analyses also had several limitations. First, the pre-post design of the knowledge assessment at baseline may have overestimated the effectiveness of health education due to potential learning effects from repeated testing. Although the research officer did not provide correct answers after the baseline assessment, the process may have primed participants to pay closer attention during subsequent education sessions, particularly since they were told while providing informed consent that they would be tested again later. However, since post-education knowledge scores remained stable over time, we believe any priming effect was minimal and likely limited to short-term score inflation immediately after client education. Loss to follow-up may have biased analyses of knowledge retention and its association with adherence in unpredictable ways.

Our study was likely underpowered to detect associations between knowledge constructs and treatment success, another key outcome. Further, our measure of nonadherence relied on self-report and may have been influenced by social desirability bias, with participants potentially underreporting missed or late doses. However, this bias would have affected our results only if misclassification differed by knowledge level, an unlikely scenario – although one that could lead to either under- or over-estimation of the association between knowledge and nonadherence.

While our study focused on knowledge and a few other individual-level predictors of nonadherence and treatment success, additional determinants at both the individual (e.g., stigma, self-efficacy, perceived social support, concurrent HIV treatment) and health-system levels may also play a significant role. Future research should explore these factors to better understand how they interact with knowledge in influencing adherence and treatment outcomes. Finally, we did not

formally measure the time required for individual or group education sessions or estimate the incremental staff costs associated with delivering structured TB education. This data gap limits our ability to comment on resource implications and cost-effectiveness, which are increasingly important considerations for TB programs operating under funding constraints.

## Conclusion

Our study found that routine education increased knowledge, and that improved knowledge was associated with favorable early treatment outcomes at a high-volume TB clinic in Kampala. However, most clients failed to achieve the minimum knowledge threshold for achieving a minimum acceptable level of TB literacy. Given these findings, the critical role of adherence in curing TB, and the likely modest cost of targeted improvements to client education, there is a need to develop and evaluate practical, scalable education strategies to enhance client knowledge and improve person-centered TB care.

## Supporting information

**S1 Appendix. Participant Questionnaire.**
(PDF)

**S1 File. Uganda NTLP TB Education Flipchart.**
(PDF)

**S2 Appendix. Individual Question Scores.**
(PDF)

## Author contributions

**Conceptualization:** Tyler Scott Johnson, Amanda J Gupta, Mari Armstrong-Hough, Irene Ayakaka, Sheela V. Shenoi, J. Lucian Davis.

**Data curation:** Tyler Scott Johnson, Joseph M. Ggita, Irene Ayakaka.

**Formal analysis:** Tyler Scott Johnson, Amanda J Gupta, Joseph M. Ggita, Mari Armstrong-Hough, J. Lucian Davis.

**Funding acquisition:** Tyler Scott Johnson, Sheela V. Shenoi, J. Lucian Davis.

**Investigation:** Tyler Scott Johnson, Leah Nanziri, Amanda J Gupta, Joseph M. Ggita, Irene Ayakaka, J. Lucian Davis.

**Methodology:** Tyler Scott Johnson, Irene Ayakaka, J. Lucian Davis.

**Project administration:** Tyler Scott Johnson, Leah Nanziri, Amanda J Gupta, Irene Ayakaka, Achilles Katamba.

**Supervision:** Amanda J Gupta, Mari Armstrong-Hough, Irene Ayakaka, Achilles Katamba, J. Lucian Davis.

**Validation:** Tyler Scott Johnson.

**Visualization:** Tyler Scott Johnson.

**Writing – original draft:** Tyler Scott Johnson.

**Writing – review & editing:** Tyler Scott Johnson, Amanda J Gupta, Joseph M. Ggita, Mari Armstrong-Hough, Sheela V. Shenoi, Achilles Katamba, J. Lucian Davis.

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
