## [Decision Letter · Decision Letter 0]

29 Oct 2025

PONE-D-25-37051Efficacy of routine tuberculosis education in Kampala, Uganda: A prospective observational cohort studyPLOS ONE

Dear Dr. Davis,

Thank you for submitting your manuscript to PLOS ONE. After careful consideration, we feel that it has merit but does not fully meet PLOS ONE’s publication criteria as it currently stands. Therefore, we invite you to submit a revised version of the manuscript that addresses the points raised during the review process.

Overall, reviewers found the manuscript well written, relevant, and ethically sound but identified several key areas for clarification and improvement. The most significant issues concern **terminology consistency and study design alignment** —authors should replace “efficacy” with “effectiveness” throughout and ensure that objectives, methods, analyses, and conclusions align accordingly. **Conceptual clarity** is needed around definitions of “TB knowledge,” distinguishing overall knowledge from disease- and treatment-specific domains, and explaining how counselling fits within an education-focused intervention. Reviewers also request **greater transparency in methods and analyses** , including justification for the sample size, details on how TB knowledge scores were calculated and weighted, clarification of statistical assumptions, and presentation of results according to objectives. **Visual and structural improvements** (figures showing study timing and knowledge change, clearer tables, and possible inclusion of flipchart images) were encouraged. Additional comments highlight the need to describe **language adaptation of materials** , **factors influencing delivery modes** , **instrument reliability** , and **data sharing compliance** . Finally, reviewers recommend discussing the **feasibility, scalability, and potential cost implications** of implementing structured TB education within routine care.

We look forward to receiving your revised manuscript.

Kind regards,

Meredith Blair Brooks

Academic Editor

PLOS ONE

Journal Requirements:

5. Please amend your authorship list in your manuscript file to include author Tyler Scott Johnson.

6. Please amend the manuscript submission data (via Edit Submission) to include author Tyler Johnson.

Additional Editor Comments :

Overall, reviewers found the manuscript well written, relevant, and ethically sound but identified several key areas for clarification and improvement. The most significant issues concern terminology consistency and study design alignment—authors should replace “efficacy” with “effectiveness” throughout and ensure that objectives, methods, analyses, and conclusions align accordingly. Conceptual clarity is needed around definitions of “TB knowledge,” distinguishing overall knowledge from disease- and treatment-specific domains, and explaining how counselling fits within an education-focused intervention. Reviewers also request greater transparency in methods and analyses, including justification for the sample size, details on how TB knowledge scores were calculated and weighted, clarification of statistical assumptions, and presentation of results according to objectives. Visual and structural improvements (figures showing study timing and knowledge change, clearer tables, and possible inclusion of flipchart images) were encouraged. Additional comments highlight the need to describe language adaptation of materials, factors influencing delivery modes, instrument reliability, and data sharing compliance. Finally, reviewers recommend discussing the feasibility, scalability, and potential cost implications of implementing structured TB education within routine care.

Reviewers' comments:

Reviewer's Responses to Questions

**Comments to the Author**

1. Is the manuscript technically sound, and do the data support the conclusions?

Reviewer #1: Yes

Reviewer #2: Partly

Reviewer #3: Yes

Reviewer #4: Yes

2. Has the statistical analysis been performed appropriately and rigorously? 

Reviewer #1: Yes

Reviewer #2: No

Reviewer #3: Yes

Reviewer #4: Yes

3. Have the authors made all data underlying the findings in their manuscript fully available?

Reviewer #1: Yes

Reviewer #2: Yes

Reviewer #3: Yes

Reviewer #4: Yes

4. Is the manuscript presented in an intelligible fashion and written in standard English?

Reviewer #1: Yes

Reviewer #2: Yes

Reviewer #3: Yes

Reviewer #4: Yes

5. Review Comments to the Author

Reviewer #1: Very well written manuscript. However, a few modifications would be of added value. 1. A simple arrow diagram showing timing of education delivery and points of knowledge assessment would be useful to visually summarize the process.

2. It would be good to know if the use of flipchart for education sessions and the use of individual vs group counselling, had any impact on the knowledge scores.

3. A visual depiction of changing knowledge scores across time would be useful.

4. The flipbook cited is in English. Is it used in same language in Kampala or are there any regional languages? In case yes, was the flipbook available in regional language?

Reviewer #2: the study conclusions are hinged on the sample size computed based on one of the objectives. it is unclear whether all objectives especially the TB knowledge increase would be evaluable. Hence the statistical approach needs a further review.

Reviewer #3: This manuscript presents a well-conducted prospective observational cohort study evaluating the effectiveness of routine tuberculosis (TB) education on patient knowledge and adherence in Kampala, Uganda. The research question is relevant, the design appropriate, and the manuscript consistent with PLOS ONE’s scope, which values methodological rigor and data transparency over novelty.

The article is technically sound and ethically compliant. The writing is clear, and the authors present results in a balanced and cautious manner. Minor revisions would enhance clarity, reproducibility, and alignment with PLOS ONE reporting standards.

1. Structure and presentation

The paper is well organized. The Abstract correctly reports the study design, sample size, and key findings. The Introduction provides appropriate background and justification. The Methods are clear and well referenced; a short statement linking the selected knowledge domains to WHO TB education frameworks would add conceptual strength. Tables and figures are relevant and clearly labeled; Table 3 could benefit from slightly improved formatting of headings and footnotes. The Discussion is well written but could emphasize implications and scalability rather than restating results.

The English is fluent and professional. Minor stylistic improvements (shorter sentences, consistent terminology, uniform abbreviations) would further improve readability.

2. Methodology and statistical reporting

The analytic methods are appropriate and transparently described. The use of Poisson regression is suitable for count data, though the manuscript should clarify whether model assumptions were verified. If overdispersion was tested and found acceptable, please state so explicitly. Reporting robust standard errors (sandwich estimators) would reassure readers about the stability of estimates in repeated measures.

Loss to follow up (23/80) is described; including a short comparison of baseline characteristics between completers and non-completers would address potential attrition bias. The small sample size is acknowledged—this should be explicitly linked to the wider confidence intervals observed in adherence-related analyses.

The newly developed TB knowledge instrument is an asset of the study. Including a brief description of its internal consistency (e.g., Cronbach’s alpha) in supplementary material would strengthen validity evidence.

3. Interpretation and discussion

The authors interpret results carefully, avoiding causal overstatements and recognizing limitations. The discussion effectively situates findings within the context of TB program implementation. Adding a concise reflection on the feasibility and scalability of integrating structured education into national programs would enhance applied relevance.

4. Data transparency and ethics

The study meets international ethical standards and includes appropriate approvals. To fully comply with PLOS ONE’s data-sharing policy, please ensure the anonymized dataset and analysis code are publicly available (Dryad/Zenodo) with a DOI cited in the Data Availability Statement.

5. Summary

This is a rigorous, ethically robust, and well-written study providing practical insights into patient education and adherence in TB care. With minor revisions—mainly clarifying statistical assumptions, adding instrument reliability data, and refining the presentation—it will fully meet PLOS ONE’s standards for clarity and reproducibility.

Recommendation: Accept after minor revision.

Reviewer #4: This is an interesting study examining the impact of TB education on adherence, strengthened by specifically studying domains within the larger topics of TB disease and treatment for impact. The study is clear and easy to follow. As health departments face increasing funding constraints, it would be useful to have some indication in the methods and discussion of the amount of time that was spent on individual and group education at the first visit, the follow up visits, and for each domain topic. The information could help provide an idea of staff cost added for TB education. I would also be interested in seeing a follow up study on cost-effectiveness. Lastly, images of the flipchart that was used might be of interest to readers as well, and if possible, to provide in the appendix.

6. PLOS authors have the option to publish the peer review history of their article (what does this mean? ). If published, this will include your full peer review and any attached files.

**Do you want your identity to be public for this peer review?** For information about this choice, including consent withdrawal, please see our Privacy Policy .

Reviewer #1: **Yes:** Swathi Krishna Njarekkattuvalappil

Reviewer #2: No

Reviewer #3: **Yes:** Robert Arana Narvaez

Reviewer #4: No

---

## [Author Response · Author response to Decision Letter 1]

26 Dec 2025

December 22, 2025

Dr. Meredith Blair Brooks

Dear Dr. Brooks,

We would like to thank the reviewers for their thoughtful review of our manuscript entitled “Effectiveness of routine tuberculosis education in a high-burden setting: A prospective observational cohort study” (PONE-D-25-37051) (title changed from “Efficacy of routine tuberculosis education in Kampala, Uganda: A prospective observational cohort study” in response to reviewer comments).

In the revised manuscript, our edits focused on three overarching priorities:

1. Improving the clarity and robustness of our analytical approach, including adding more detail about our power calculations, filling in the complete treatment outcome results, presenting results centered around overall knowledge scores, and shifting our emphasis to the proportion of participants who achieved TB literacy rather than reporting knowledge increases without clarifying whether those increases were sufficient.

2. Strengthening the clarity of our definitions of key concepts, including refining our use of “effectiveness” versus “efficacy,” distinguishing “education” from “education + counseling,” clarifying how and why we defined overall knowledge versus knowledge constructs, and specifying our definition of TB literacy as achieving >90% for a concept.

3. Enhancing clarity in study procedures, including adding a figure to show the time points of education delivery and knowledge assessment, and incorporating the flip book into the supplemental material.

We also made several additional revisions based on specific reviewer feedback not included in the numbered comments, including:

● We added a new figure, Figure 1, to illustrate the timing of education delivery and the points of knowledge assessment (Lines 119).

● We added a discussion of programmatic implications (Lines 306-313) and acknowledged a lack of time and cost data as a study limitation (Lines 342-343).

● We accounted for mild overdispersion in the Poisson regression model, adjusting our results accordingly and noting this in the Methods (Lines 150-151).

● We addressed the concern that loss to follow-up could lead to attrition bias by comparing participants lost to follow-up with those with a recorded final treatment outcome on key baseline characteristics; we did not find significant differences between the groups. This new analysis is noted in the Methods (Lines 156-159) and Results (Lines 252-254).

● We replaced our original figure with a new one that focuses on the proportion of individuals who achieved TB literacy (Lines 219-220).

Please see below for a point-by-point response to each Reviewer's numbered comments, including references to the line numbers in the revised text where we have implemented the requested changes.

Abstract

Comment 1: In the Abstract, the authors have an inconsistency between the title of the manuscript (“efficacy”) and the information in the background, which refers to effectiveness (Line 4). I recommend the authors stick with effectiveness that best suits the methodology of this study.

Response: Thank you for this suggestion. We agree and have updated the title accordingly. We now describe changes in TB knowledge after routine education as a measure of effectiveness rather than as a measure of efficacy.

Comment 2: The abstract title is about education, but the authors have included counselling in line with a different concept. I recommend the authors stick to the education aspect in order not to confuse the readers.

Response: Thank you for highlighting this distinction. We have removed all references to “education and counseling” and now focus solely on “education,” as there is no standardized guideline or approach to counseling in routine TB care in Uganda. We have also added definitions for education and counseling in Lines 51-55 on Page 5 to clarify this distinction.

Introduction

Comment 3: The authors explain the gaps in the current approach and propose to explore studying the specific TB knowledge aspects influenced by education in the routine setting. The authors however again refer to “efficacy” of intervention in the routine setting in Line 46. In routine setting, the authors could not control study events, the appropriate term should be “effectiveness.”

Response: As noted in our response to Comment 1, we have revised the manuscript to replace all references to ‘efficacy’ with ‘effectiveness.’

Materials and Methods

Comment 4: The authors revise appropriately to replace the wording of efficacy with effectiveness in line 53, 60

Response: As noted in our response to Comment 1, we have revised the manuscript to replace all references to ‘efficacy’ with ‘effectiveness.’

Comment 5: Primary Objective 1 focuses on TB knowledge. The authors have defined TB knowledge to encompass TB disease and treatment constructs. According to Line 38 it suggests that TB knowledge is a combination of TB disease and TB treatment. The authors further define TB specific knowledge constructs. This creates confusion since the conclusions focus on the TB specific knowledge constructs. I recommend that the authors clearly differentiate between these two concepts and report on results aligned with the TB knowledge as stated in the objectives .

Response: Thank you for this suggestion. To address the Reviewer’s concern, we now focus our analyses on overall TB knowledge, as measured by scoring the entire TB knowledge instrument as explained in Lines 127-131 on Page 8:

“To score the knowledge instrument, we calculated the proportion of correct responses to questions related to each construct for each participant at each time point to create eight construct-specific scores (individual question scores included in S2 Appendix). We then averaged these construct scores to create an overall TB knowledge score and standardized the resulting value on a 0-100 scale.”

In addition, we have removed analyses of “treatment-specific” and “disease-specific” knowledge constructs, while retaining our secondary analyses examining the effect of education on eight content-specific constructs, as described below (see Lines 141-142, Page 9):

“We assessed changes in TB knowledge after baseline client education and evaluated knowledge retention at each follow-up visit using two-sided paired t-tests overall and at the construct level.”

Development of an instrument for assessing TB knowledge

Comment 6: The authors have explained how they developed the tool and provided the reference materials. I however note that the authors considered adding counselling to the tool whereas the objectives are focused on TB education. The authors should clarify to what extent counselling was conducted and how that influenced the TB education for the clients.

Response: Thank you for the question. Please see our complete response contained within Comment 2.

Comment 7: I have reviewed the S1 appendix and found that it contains basic questions understandable to the clients. The authors, however, are using complex terms such as pathogenesis, treatment mechanisms etc which are not a representation of the S1 Appendix nor the description or expectation of the client to be given to the clients. I recommend authors use terms consistent with the questions asked such as for TB disease (TB disease, TB symptoms, TB transmission, TB prevention, TB/HIV relationship) and for TB treatment (TB medicines, how to take TB medicines, Response to TB treatment, TB treatment follow up visits, adherence to treatment).

Response: Thank you for this suggestion. We agree that the previous construct names did not reflect the simplicity of the questions and have renamed them accordingly (Lines 85-87, Page 6). We have also updated the S1 Appendix so that each questionnaire item links to the updated construct name.

Procedures

Comment 8: The authors explain how the education was conducted. The method of delivery was not the same and the materials used to deliver were not the same for all the participants. For clarification to the readers, the authors should explain the factors that determined the different delivery methods and discuss their impact on the results of this study.

Response: Thank you for the question. Because our goal was to evaluate routine TB education, we described the national guidelines on TB education in Lines 73-75 on Page 6. Since these guidelines give clinicians substantial discretion in how they deliver education, we have added a clarifying statement for readers in Lines 78-79 on Page 6:

The content and delivery of TB education lack standardization and are primarily left to individual staff. At Kisenyi, TB unit nurses and assistant nurses provide client education individually or in groups, sometimes using the flipchart to illustrate key concepts.

Table 1 presents the frequency of use of a standardized educational aid (the TB Education Flipchart) and the education modality used (individual vs. group). We also include the education modality as a covariate in both the Poisson and logistic regression models (Table 3).

Sample size estimates

Comment 9: It is not clear why the authors did not provide or make a statement of why the objective of education to increase TB knowledge was not used to compute sample size. It is important that the readers are sure about the power to make the conclusions. The authors indicate TB knowledge is a combination of TB disease and TB treatment knowledge. The authors should demonstrate and justify that the sample size used was the highest and therefore provides power for all objectives. I have reviewed the references for sample size assumptions and find no clear linkage. Since sample size is critical for validity of conclusions, I recommend that the authors revisit the sample size computation approach and assure the readers about the validity of the conclusions.

Response: We appreciate the Reviewer’s question. While we conducted power calculations for all four objectives a priori, we included in our initial submission only the calculation for the analysis with the least power (Objective 3). We have amended the sample size section to detail how power calculations were conducted for each of the four objectives (Lines 58-62, Page 4). We also reported 95% confidence intervals around each effect estimate as measures of uncertainty.

Statistical analysis

Comment 10

The authors should provide the scientific merit of weighting responses to questions. I think a client would have knowledge or not to a particular question. For the interest of the readers, the authors should provide the scores for the questions in S1 Appendix and provide how the percentage score for TB knowledge was determined for each client. It is not clear why the statistical analysis was not focused on the TB knowledge increase as the outcome but rather the nonadherence. This approach confuses the readers and makes it difficult to relate the study objectives with the results and conclusions. The authors should provide more clarity how the scores were generated and how they are related to the percentages.

Response: Thank you for these questions. As shown in the S1 Appendix, some constructs required multiple questions to assess, while others had only a single question. We’ve added text to clarify the calculation of the 8 construct scores and their equal weighting in the overall TB score in Lines 127-131, Page 8.

We have also added a new appendix, S2 Appendix, listing the mean scores for each question across all measurement time points.

As detailed in our response to Comment 5, we have redefined our primary measure of TB knowledge as the overall TB score, which now serves as the primary outcome for Objective 1 (Effect of education on TB knowledge at baseline) and Objective 2 (Effect of education on TB knowledge over time). In contrast, Objective 3 (Effect of post-education TB knowledge on nonadherence) and Objective 4 (Effect of post-education TB knowledge on treatment success) examine TB knowledge as a predictor.

Results

Comment 11: Line 166-195; This section on results is unclear. the authors should display the results for easy understanding and following of the readers. I recommend the results are presented in accordance with the objectives presented. The disease specific aspects are secondary results and should be provided as such. The clients reaching adequate knowledge are not shown in the table and yet this is central to the study. I recommend these results are clearly shown to the readers.

Response: Thank you for these comments. As detailed in our responses to earlier comments above, we have redefined our primary measure of TB knowledge as the overall TB knowledge score.

Regarding adequate TB knowledge, we have introduced a new term, “TB literacy,” in Lines 131-135, Page 8:

To enhance interpretability, we defined a post-hoc secondary outcome, TB literacy, operationalized as an overall TB knowledge score ≥90 points.

We have replaced our initial Figure 2 with a new figure showing the proportion of clients who achieved TB literacy overall and for each construct.

Comment 12: Line 211-225; The authors intend to report on TB knowledge and nonadherence in this section. There is no result on TB knowledge and nonadherence instead of the authors report on the domains. This is inconsistent with the set-out objectives and consequently lack of power to make any valid conclusions. I recommend the authors first display results aligned with objectives and can then display other sub analysis results.

Response: Thank you again for highlighting this distinction. As stated above in responses to earlier Reviewer Comments, especially Comment 5, we have changed our measure of TB knowledge to the overall TB knowledge score, which aligns with the approach stated in the objectives.

Comment 13: Line 235- 242; the authors intend to report on the association between TB knowledge and treatment outcomes. The authors, however, selectively report of a few domains. The overall TB knowledge should be the predictor variable rather than some domains. Since the study was not powered to conclude on association between the type of education delivery method and outcome, it is for the authors to acknowledge lack of power to make the conclusion.

Response: Thank you for this suggestion. As stated in our above response to Comment 10, we have changed the predictor measure for TB knowledge for Objectives 3 and 4 to the overall TB knowledge score. Adjusted measures of association at each time point for these Objectives are now provided in an updated Table 3.

Comments 14: Line 304-323; the authors have stated the limitation to this study. However, uncertainty about the power calculation based on nonadherence while the primary study was TB knowledge increase should be made clear so the readers put the results of this work in proper context.

Response: Yes, of course. As noted in our response to Comment 9, we have provided power and sample size calculations for all four objectives and report 95% confidence intervals around each effect estimate as measures of uncertainty.

Comment 15: Line 326; the conclusion statement implies that routine education sustains knowledge. This is not what the methodology performed. Routine education was only provided at baseline and therefore there is no evidence to suggest it sustained the knowledge. I recommend the authors revise the sentence.

Response: Thank you for this. We have added a sentence to the Conclusion to address this comment (found on Lines 346-348, Page 21):

Our study found that routine education increased knowledge, and that improved knowledge was associated with favorable early treatment outcomes at a high-volume TB clinic in Kampala.

Sincerely,

J. Lucian (Luke) Davis, MD, MAS

Associate Professor

Yale School of Public Health & Yale School of Medicine

---

## [Decision Letter · Decision Letter 1]

17 Feb 2026

Effectiveness of routine tuberculosis education in in a high-burden setting: A prospective observational cohort study

PONE-D-25-37051R1

Dear Dr. Davis,

We’re pleased to inform you that your manuscript has been judged scientifically suitable for publication and will be formally accepted for publication once it meets all outstanding technical requirements.

Kind regards,

Hamufare Dumisani Mugauri, Ph.D. Epidemiology and Public Health

Academic Editor

PLOS One

Reviewers' comments:

Reviewer's Responses to Questions

**Comments to the Author**

1. If the authors have adequately addressed your comments raised in a previous round of review and you feel that this manuscript is now acceptable for publication, you may indicate that here to bypass the “Comments to the Author” section, enter your conflict of interest statement in the “Confidential to Editor” section, and submit your "Accept" recommendation.

Reviewer #2: All comments have been addressed

Reviewer #3: All comments have been addressed

2. Is the manuscript technically sound, and do the data support the conclusions?

Reviewer #2: Yes

Reviewer #3: Yes

3. Has the statistical analysis been performed appropriately and rigorously? 

Reviewer #2: Yes

Reviewer #3: Yes

4. Have the authors made all data underlying the findings in their manuscript fully available?

Reviewer #2: Yes

Reviewer #3: Yes

5. Is the manuscript presented in an intelligible fashion and written in standard English?

Reviewer #2: Yes

Reviewer #3: No

6. Review Comments to the Author

Reviewer #2: (No Response)

Reviewer #3: The manuscript is close to publishable and the main message is clear: routine TB education improves TB knowledge in the short term, but sustained “TB literacy” remains uncommon. Before acceptance, I recommend the following minor but essential revisions:

Mandatory editorial correction: The title contains an obvious duplication (“in in a high-burden setting”). Please correct this and complete a final language/consistency pass (PLOS ONE does not copyedit accepted manuscripts).

Intervention clarity and consistency: Please ensure consistent terminology throughout (e.g., “education” vs “counseling”) and provide a brief, standardized description of the minimum intervention components (core topics covered, who delivers it, and approximate duration).

Cautious interpretation: When discussing clinical end outcomes, keep conclusions appropriately cautious and aligned with the programmatic/observational nature of the study; avoid strong causal wording.

Key limitations: Please explicitly emphasize that adherence was self-reported and may be affected by recall/social desirability bias, and that loss to follow-up may influence estimates of knowledge retention.

Recommendation: Minor revision.

7. PLOS authors have the option to publish the peer review history of their article (what does this mean? ). If published, this will include your full peer review and any attached files.

**Do you want your identity to be public for this peer review?** For information about this choice, including consent withdrawal, please see our Privacy Policy .

Reviewer #2: **Yes:** eric wobudeya

Reviewer #3: No

---

## [Editor Report · Acceptance letter]

PONE-D-25-37051R1

PLOS One

Dear Dr. Davis,

I'm pleased to inform you that your manuscript has been deemed suitable for publication in PLOS One. Congratulations! Your manuscript is now being handed over to our production team.

Kind regards,

on behalf of

Dr Hamufare Dumisani Mugauri

Academic Editor

PLOS One